

# Spatiotemporal dynamics of soil phosphorus and crop uptake in global cropland during the twentieth century

Jie Zhang[1], Arthur H.W.Beusen[2,3], Dirk F.van Apeldoorn[2,4], José M. Mogollón[2], Chaoqing Yu[1], Alexander F. Bouwman[2,3]

[1]Department of Earth System Science, Tsinghua University, Beijing, 100084, China

[2]Department of Earth Sciences-Geosciences, Faculty of Geosciences, Utrecht University, P.O.Box 80021, 3508 TA Utrecht, The Netherlands

[3]PBL Netherlands Environmental Assessment Agency, P.O.Box 30314, 2500 GH The Hague, The Netherlands

[4]Department of Plant Sciences, Wageningen University, P.O.Box 430, 6700 AK Wageningen, The Netherlands

*Correspondence to*: Chaoqing Yu (chaoqingyu@tsinghua.edu.cn); A. F. Bouwman (a.f.bouwman@uu.nl)

**Abstract.** Phosphorus (P) plays a vital role in global crop production and food security. In this study, we investigate the changes in soil P pools and crop P uptake, using a 0.5 by 0.5 degree spatially explicit model for the period 1900-2010. The simulated country-scale crop P uptake agrees well with historical P uptake. Simulated crop P uptake is influenced by both soil properties (available P and the P retention potential) and crop characteristics (maximum uptake). Until 1950, P fertilizer application had a negligible influence on crop uptake, but recently it has become a driving factor for food production in industrialized countries and a number of transition countries like Brazil, Korea and China. Globally, the total P pool per hectare increased rapidly between 1900 and 2010 in soils of Europe (+31%), South America (+2%), North America (+15%), Asia (+17%) and Oceania (+17%), while it has been stable in Africa. This comprehensive and spatially explicit model can be used to assess the dynamics of soil P inventories, which serve as indicators for soil fertility and productivity.

**Keywords:** cropland; crop uptake; dynamic; fertilizer; global; manure; modelling; phosphorus; soil reserves





# 1 Introduction

The current world population of 7.3 billion is expected to reach 9.7 billion in 2050 (UN, 2016), likely triggering a greater demand for food and resources. Moreover, increasing prosperity will lead to further shifts in human diets towards more meat and milk consumption, particularly in developing countries (Alexandratos and Bruinsma, 2012). Livestock products require

more nutrients for production compared to crops and thus may induce additional nutrient demands (Bouwman et al., 2013).

Phosphorus (P) is one of the major limiting nutrients in agriculture (Koning et al., 2008), which unlike nitrogen, cannot be fixed from the atmosphere by living organisms or industrial processes. In early agricultural systems, P was supplied to soils by recycling animal manure, crushed animal bones, human excreta, city waste and ash (Beaton, 2006). Since the industrial

revolution, however, soil P enrichment has been increasingly dominated by non-renewable P resources such as guano (accumulated seabird droppings) and phosphate rock. Presently, phosphate rock provides the P for producing 90% of global P fertilizer use (PotashCorp, 2016), which was 18.8 Tg P (Tg = teragram; 1 Tg = $10^{12}$g = 1 million metric tons) in 2010 (FAO, 2016a).Yet, this resource is rapidly dwindling with estimates for phosphate rock availability ranging from peak P production in 2033 with subsequent rapid decline (Cordell et al., 2009) to half of the resources being used by 2100 (Van

Vuuren et al., 2010) or complete exhaustion within the next 300-400 years (Van Kauwenbergh, 2010).

The availability of P to plant roots is determined by the concentration of phosphate ions in the soil solution and the ability of the soil to replenish them after plant uptake (Syers et al., 2008). The replenishing of phosphate ions depends on soil characteristics such as mineralogy, soil reaction and degree of weathering (Fairhurst et al., 1999). Soil materials rich in

soluble alumina or iron, a high calcium activity, or clay minerals like kaolinite react with P to form insoluble compounds inaccessible to plant roots (Brady, 1990). Any surplus P application over crop uptake and losses by runoff and erosion accumulates in the soil as "residual P" and it can be available for crop uptake for many years depending on the soil characteristics and management (Syers et al., 2008;Batjes, 2011). Ideally, good crop yields may be maintained when adequate P is present in the readily available pools to maintain a critical threshold P concentration in the soil solution (Syers

et al., 2008) and with annual P inputs from fertilizer that compensate plant P uptake and erosion loss. However, when the amount of readily available P is below this critical threshold level, the rate of P release from residual P is insufficient to sustain optimal crop yields. Improving our mechanistic understanding of soil P dynamics locally and globally is important for evaluating broad-scale food security and improving sustainable P management.

This paper presents a global, spatially explicit analysis of the historical P uptake in global crop production systems using a soil model with two P pools (Wolf et al., 1987;Janssen et al., 1987), which was later presented as the "Dynamic Phosphorus Pool Simulator" (DPPS)(Sattari et al., 2012). DPPS describes the impact of long term P application on the transfers between a stable and a labile soil P pool as well as crop P uptake, accounting for weathering, deposition and erosion (Fig. 1). It was





first developed for the field scale (Wolf et al., 1987;Janssen et al., 1987) and has recently been applied to simulate the impact of soil residual P on crop uptake and future demand for fertilizer P on the continental scale (Sattari et al., 2012). The aim of this paper is to develop a spatially explicit version of the DPPS model, and to simulate the soil P dynamics, soil P fertility and crop P uptake for global cropland from the beginning of the 20th century up until the year 2010.

## 2 Material and methods

### 2.1 Model description

The spatially explicit DDPS model presented here describes the dynamics of P uptake by plants as a function of two soil P pools, i.e. labile soil P pool ($LP$, kg P ha$^{-1}$) and stable soil P pool ($SP$, kg P ha$^{-1}$) with an annual temporal scale and a spatial resolution of 0.5 by 0.5 degree (Fig. 1). $LP$ represents all forms of P that can directly be taken up by plant roots, comprising both organic and inorganic P; $SP$ represents forms of P bound to soil minerals and organic matter that are not directly available to plants (Table 1). P is transferred in both direction between two pools.

Our model considers natural occurring inputs to the soil, i.e. weathering ($weathering$, kg P ha$^{-1}$yr$^{-1}$), litter ($litter$, kg P ha$^{-1}$yr$^{-1}$) and atmospheric deposition ($deposition$, kg P ha$^{-1}$yr$^{-1}$); and anthropogenic P inputs, including application of mineral P fertilizer ($fertilizer$, kg P ha$^{-1}$yr$^{-1}$) and animal manure ($manure$, kg P ha$^{-1}$yr$^{-1}$). P outflows from the soil system include the withdrawal of P in harvested crops ($uptake$, kg P ha$^{-1}$yr$^{-1}$) and runoff ($runoff$, kg P ha$^{-1}$yr$^{-1}$). Figure 1 shows a schematic diagram of the P pools, fluxes, inputs and outputs that comprise the model and Table 1 shows the parameters of the DPPS model.

The thickness of the topsoil is assumed constant at 30cm by fresh subsoil material (>30 cm) ($fresh\_soil$, kg P ha$^{-1}$yr$^{-1}$) compensating the topsoil lost by runoff. The proportion of $LP$ and $SP$ in $fresh\_soil$ is based on the pools according to the soil P inventory depicted in Yang et al. (2010). The P input from $litter$ (kg P ha$^{-1}$yr$^{-1}$) remaining after crop harvest is returned to the soil and becomes part of the $LP$ pool. For natural ecosystems, assuming there is no anthropogenic fertilizer application, the uptake equals the inputs from $litter$, $deposition$ and $weathering$. In contrast, P $uptake$ in agricultural soils is calculated explicitly (see below).

P from atmospheric deposition obtained from the global results of Model of Atmospheric Transport and Chemistry (MATCH) is assumed to be a direct input for the SP pool only, since mineral aerosols (dust) are the dominant source of atmospheric P (~ 82%)(Mahowald et al., 2008) and are not readily available for plant uptake. P from weathering in global cropland is assumed to amount to 1.6 Tg yr$^{-1}$ (Liu et al., 2008). This value is used to calculate the weathering fraction of the soil P in apatite ($fr\_weathering$ = 0.001747) from the global natural soil P inventory by Yang et al. (2013), and assumed to be available directly for plant uptake.




P runoff is calculated following the approach of Beusen et al. (2015).The P content of the soil loss (*runoff_LP* and *runoff_SP*) is based on the *LP* and *SP* of the soil in the grid cell considered at that moment in time.

The yearly changes of the *LP* and *SP* pools (kg P ha$^{-1}$yr$^{-1}$) are calculated as follows:

$$\frac{\partial_{LP}}{\partial_t} = F_{sp2lp} - F_{lp2sp} + weathering + fertilizer + manure + freshsoil_{LP} + litter - runoff\_LP - uptake \quad (1)$$

$$\frac{\partial_{SP}}{\partial_t} = F_{lp2sp} - F_{sp2lp} + deposition + freshsoil_{SP} - runoff\_SP \quad (2)$$

Where *Flp2sp* and *Fsp2lp* (kg P ha$^{-1}$yr$^{-1}$) are the fluxes between *LP* and *SP* and *SP* to *LP*, respectively:

$$F_{lp2sp} = \frac{LP}{\mu_{LS}} \quad (3)$$

and:

$$F_{sp2lp} = \frac{SP}{\mu_{SL}} \quad (4)$$

where the variables $\mu_{LS}$ and $\mu_{SL}$ are transfer times (years) between *LP* to *SP* and *SP* to *LP*, respectively. The quotients $\frac{1}{\mu_{LS}}$ and $\frac{1}{\mu_{SL}}$ are the transfer rates (year$^{-1}$). The transfer time $\mu_{SL}$ from *LP* to *SP* is set to 5 years based on the original DPPS (Janssen et al., 1987). The transfer time from *SP* to *LP* ($\mu_{SL}$) is calculated for every grid cell based on the mass

balance of *LP* for natural ecosystems using Eq. (1) and assuming steady state:

$$\mu_{SL} = \frac{SP}{LP \times \frac{1}{\mu_{LS}} + deposition} \quad (5)$$

We selected the mass balance of *LP* (Eq. 1) to calculate $\mu_{SL}$ because it yields a value that matches the value obtained by Wolf et al. (1987) based on experimental data.

The amount of P that can be accessed by plant roots (*availableP*, kg P ha$^{-1}$yr$^{-1}$) is calculated as a fraction of *LP* (*fr_mobile×LP*). The P uptake by crops was calculated using Michaelis-Menten kinetics (Michaelis and Menten, 1913;for elaboration for nutrient uptake see Nijland et al., 2008) as follows:

$$uptake = \frac{max\_uptake \times availableP}{\left(\frac{c \times max\_uptake}{init\_recovery} + availableP\right)} \quad (6)$$

Where *max_uptake* (kg P ha$^{-1}$yr$^{-1}$) is the maximum P uptake, and *init_recovery* is the initial recovery fraction (no dimension),

which is the initial slope of the P response curve presented (Batjes, 2011) for all soil types distinguished in the legend of the FAO-Unesco soil map of the world (FAO-Unesco, 1974); *c* is a constant to obtain the *availableP* for which uptake is 0.5





times *max_uptake* (no dimension; *c* = 0.5). The calculation of the area-weighted value of *init_recovery* for each grid cell is based on the sub-grid distribution of soil classes.

*Max_uptake* is a time dependent variable, considering the development of technology in crop production. Its absolute
5   maximum value is 100 kg P ha$^{-1}$ yr$^{-1}$, which exceeds the present highest value for all countries of the world based on our data (see Sect. 2.2). For countries with no cropland, *max_uptake* is set at a value of 5 kg P ha$^{-1}$ yr$^{-1}$ for 1900 to 1950 and 10 kg P ha$^{-1}$ yr$^{-1}$ after 1950. For countries with cropland, before 1960 *max_uptake* is the minimum of historical uptake and the sum of the P inputs from manure and fertilizers times 2; after 1960, it is the historical uptake times a factor of 2. It should be noted that *max_uptake* won't decrease in countries with cropland.

In grid cells where cropland expansion takes place, the initial conditions of virgin soil (without fertilizer history) are assigned to the new, additional area in the year considered. For land abandonment (arable land to natural land) we assume that it takes 30 years for abandoned land to revert to natural conditions, and in this period the P in *litter* increases and *uptake* decreases linearly with time from zero to the natural flux.

For the comparison between model and data on P uptake, we use the root mean square error (*RMSE*) (see S1).The model sensitivity is investigated using Latin Hypercube Sampling (LHS), with uncertainty ranges for 12 model parameters (Table 2), respectively, and expressed as the standardized regression coefficient (*SRC*), to compare model output for global P uptake. A detailed description of the LHS approach is provided in S2. In the sensitivity analysis we focused on the modelled *LP* and
*SP* size, and crop P *uptake* in 1950 and 2000.

To initialize the model in 1900, we assume that the pools are in equilibrium and the sum of *LP* and *SP* is equal to the total P (*TP*) in natural systems as defined by Yang et al. (2013). Each cell is initialized in 1900 with *LP* and *SP* from the global gridded soil P inventory (Yang et al., 2013), representing the pre-industrial conditions. The model is calibrated by varying
*fr_mobile* to achieve a good fit with the crop uptake data for each country. This calibration is an iterative procedure that estimates the error (kg P ha$^{-1}$yr$^{-1}$) in the calculated uptake:

$$error = \frac{\sum_1^n uptake_{model} - uptake_{data}}{n} \tag{7}$$

where *n* is the number of data points between 1960 and 2010. We compared every second year, so *n* = 26. *fr_mobile* was limited to the range [0.01, 0.7]. The calibration is considered successful when ABS(*error*) < 0.4. For countries where the
calibration is not successful, we applied the regional (Table S1) average area-weighted *fr_mobile*.





## 2.2 Data used

We use the spatially explicit data on P fertilizer use and animal manure spreading, land use, crop production and cropland areas from the database used in the IMAGE-Global Nutrient Model (Beusen et al., 2016).

5   For the period 1900-1960, the gridded inventories of P inputs from a recent study (Bouwman et al., 2013) were used for the calculation of the soil P pools and crop uptake. This dataset was based on various sources for land use (Klein Goldewijk et al., 2011;Klein Goldewijk et al., 2010), livestock for 1900-1960 (Mitchell, 1998, 1993a, b), animal and crop production and fertilizer use for 1930-1950 (FAO, 1951) and fertilizer use prior to 1930 (Cressy Morrison, 1937), which were spatially distributed with IMAGE-GNM as described by Bouwman et al. (2013).

Crop P uptake and fertilizer inputs per hectare are spatially homogeneous for all grid cells with cropland within each country. The total production of P from animal manure was estimated from country livestock data for nondairy and dairy cattle, pigs, poultry, sheep and goats, and the P content in the manure (Table S2). The amount of manure available for spreading in agricultural land excludes droppings in grassland, manure used as fuel or building material or manure otherwise ending
outside the agricultural system (e.g. in lagoons) (Beusen et al., 2016).

IMAGE-GNM distinguishes two crop systems, i.e. (1) crops in mixed systems, in which there is a linkage between crop and livestock production through manure (from animals to crops) and feed (from crops to animals) exchanges, and (2) crops in pastoral systems, in which crop and livestock production are separate systems. While P fertilizers are the same in all
cropland within a country, P inputs from animal manure are different in mixed and pastoral systems.

Simulated crop uptake is validated against crop data covering all annual and perennial food, feed and fodder crops and fruits from FAO statistics (FAO, 2016b, a) and are aggregated to 34 crop groups distinguished in recent FAO studies (Alexandratos and Bruinsma, 2012). The P uptake per hectare is obtained from the crop yields and P content for each crop
group, as listed in Table S3.

We present the data and results for a number of individual countries (China, United States, South Africa and France), and world regions. The definition of the regions is provided in Table S1.

### 2.3 Sensitivity analysis

We investigate the model sensitivity of 12 parameters for two years (1950 and 2000) by using Latin Hypercube Sampling (LHS, Table 2). The sensitivity is expressed using the standardized regression coefficient (*SRC*, Table 3). A detailed description of the approach for the sensitivity analysis and the results is in Sect. S2 in the Supplement (S).



## 3 Results

### 3.1 Model performance

The DPPS model adequately simulates P uptake by crops in different regions and countries (Fig. 2). The modelled P uptake values match well the historical records (Fig. 3) as shown by the *RMSE* of 19% (26 regions, 26 year, 676 points). At the
country scale, simulations and observations of annual P uptake for every second year (1960-2010) agree well (*RMSE* = 45%, 201 countries, 26 years, 5226 points). The *RMSE* of the country-level simulation exceeds that of the aggregated results for world regions, which is caused by the underestimation of peak values by DPPS. However, for the global and regional scale we consider the DPPS results acceptable (< 50% for global scale).

### 3.2 P application and crop P uptake

The global P inputs (including mineral fertilizer and manure) have increased from 2.0 Tg P in 1900 to 23.0 Tg P in 2010 with large variations across different regions and countries of the world. Between 1900 and 2010, the global use of mineral fertilizer in croplands increased from 0.4 Tg to 15.8 Tg yr$^{-1}$ and the use of manure from 1.6 Tg to 7.2 Tg yr$^{-1}$. At the global scale, about half of the applied P in 2010 was taken up by harvested crops (modelled 12 Tg P yr$^{-1}$, 7.3 kg P ha$^{-1}$yr$^{-1}$; data 13.8 Tg P yr$^{-1}$, 8 kg P ha$^{-1}$yr$^{-1}$). All world regions and countries show similar patterns before 1950, i.e. very low P input levels and
crop P uptakes that do not differ much from the inputs.

The annual P inputs in Western Europe increased from 0.4 Tg P in 1900 to 2 Tg P in 1980 and then decreased to 0.9 Tg P in 2010, which translates into rates of 24 kg P ha$^{-1}$yr$^{-1}$ in 1980, followed by a gradually decrease to 11 kg P ha$^{-1}$yr$^{-1}$ in 2010 (Fig. 2a). Crop uptake started to rapidly increase in Western Europe from 4 kg P ha$^{-1}$yr$^{-1}$ in 1950 to 13 kg P ha$^{-1}$yr$^{-1}$ in 2010; in
2010, the crop P uptake exceeded the P fertilizer and manure application (Fig. 2a), and two levels of crop uptake have been achieved in different years at the same application rate (Fig. 4a and 4j, Western Europe and France). This indicates that since the 1980s, P application has been reduced while uptake continues to increase due to the supply of residual soil P (Fig. 5a). The model results also show this hysteresis, although DPPS underestimates the P uptake in the most recent years (Fig. 2a).

Agriculture in Western Europe is much more intensive than in the United States (Van Grinsven et al., 2015). P inputs in North America peaked in 1980 with a total of 2.9 Tg yr$^{-1}$. Application rates increased from 1 kg P ha$^{-1}$ in 1900 to 12 kg P ha$^{-1}$yr$^{-1}$ in 1980 and 10 kg P ha$^{-1}$yr$^{-1}$ in 2010, and crop P uptake lagged behind P inputs for around 20 years but inputs and uptake are now at a similar level (Fig. 5b). The trend is visible in United States data, and DPPS results agree well with the uptake based on production data in both cases (Fig. 2h and 2b).

In contrast to the other continents, the input level in Africa is much lower, with annual application rates ranging from 1 kg P ha$^{-1}$yr$^{-1}$ in 1900 to 5 kg P ha$^{-1}$yr$^{-1}$ in 2010. Along with the slowly increasing application rates, African crop uptake is also



increasing at a low rate with no accumulation of soil P (Fig. 5c). DPPS results are in good agreement with the production-based uptake for Africa (Fig. 2c and 3c).

The annual P application in Asia increased more than 12-fold from 0.9 Tg yr$^{-1}$ in 1990 to 12.4 Tg yr$^{-1}$ in 2010, and manure P

more than doubled from 0.8 Tg yr$^{-1}$ to 3.5 Tg yr$^{-1}$ from 1960 to 2010. The P application rates of Asia rose dramatically from low values in 1900 to 22 kg P ha$^{-1}$ yr$^{-1}$ in 2010. Asia is currently in the phase of rapidly increasing inputs, with crop P uptake also increasing but at a slower rate (Fig. 2d). DPPS results are in good agreement with the uptake data for this continent (Fig. 2d and 3d).

In South America, the annual P inputs increased from 5 to 23 kg P ha$^{-1}$yr$^{-1}$ between 1900 and 2010 and the annual P uptake also increased from 4 to 13 kg P ha$^{-1}$yr$^{-1}$ for the same period of time. Although DPPS underestimates the P uptake in recent years, the simulations agree with the measured data for most years (Fig. 2e and 3e).

In Oceania, annual P application varied between 10 and 13 kg P ha$^{-1}$yr$^{-1}$ in recent decades. Oceania shows low uptake rates

relative to inputs over the whole period 1900-2010, and the simulated results are in good agreement with the data (Fig. 2f and 3f). This indicates that the cumulative inputs of P fertilizer and manure exceed the crop P uptake.

### 3.3 Soil P budgets

The soil P budget is defined as the biogeochemical soil P budget, which is the difference between soil P inputs (mineral fertilizer, manure and deposition) and outputs (uptake and runoff)(Fig. 5 and 6a).

The soil budgets (Fig. 5) show the difference between the industrialized countries that are currently in an equilibrium or depletion phase with regard to soil P, transition countries that are in an accumulation phase (e.g. China, India and South America) and countries with a low input and productivity crop production system (e.g. Africa). The soil budgets also show how important runoff losses are for the budgets and for determining whether P is accumulating (Fig. 6a).

We also calculated the agronomic soil P budget, which is the difference between crop P uptake and inputs (the sum of fertilizer and manure)(Fig. S1c). The crop P uptake and soil P budget have a large spatial heterogeneity. China, India, and Brazil are among the countries with the highest P input rates in 2010 (Fig. S1a). China, USA, Eastern Africa and Western Europe are the areas with highest P uptake (Fig. S1b). However, South Africa (like Brazil), China, India and central America

30    are regions with large P surpluses. Western Europe, Western Russia and Western Africa currently show P deficits (Fig. 6a).





### 3.4 Soil P pools

The spatial distribution of the changes in the *LP*, *SP* and *TP* (*TP = LP + SP*) pools were generated yearly from 1900 until 2010. In the initial condition (1900), soils in wet tropical climates generally had low P contents as a result of prolonged weathering (Yang and Post, 2011)(Fig. S2). Nevertheless, with the advent of intensive agricultural activities during the 20[th] century and up until 2010, *TP* has drastically increased along the western coast of the Americas, central and western India, southwest part of Saudi Arabia and Ethiopia (Fig. 6c).

The total P content per hectare increased rapidly between 1900 and 2010 in soils of Europe (31% relative to 1900), North America (15%), Asia (17%) and Oceania (17%), with a small increase in South America (2%), while total P content has been stable in Africa (Fig. 7c). This increase primarily occurred in the decades after 1950.

The labile P content per hectare of Western Europe was relatively stable from 1900 to 1930 (260 to 272 kg P ha$^{-1}$), then increased from 1940 to 1980 (314 to 387 kg P ha$^{-1}$) and decreased from 1980 to 2010 (387 to 350 kg P ha$^{-1}$), due to the decreasing P fertilizer application rates in recent years. *LP* in Asia and South America has been increasing, especially since 1980. In Africa, *LP* has remained consistently stable, while more recently it has become stable in Oceania and North America (Fig. 7a).

### 4 Discussion

Sattari et al. (2012) applied the DPPS model to reproduce historical continental crop P uptake (1965-2007) and estimate P requirements for crop production in 2050 using fixed $\mu_{LS}$ and $\mu_{SL}$. Our longer simulation time (1900-2010), as well as spatially explicit calculations (0.5 by 0.5 degree resolution) allows to assess long-term changes in soil P pools taking into account local heterogeneity. Further improvements include the consideration of yearly land use changes, spatially explicit soil properties and P contents, erosion and runoff, the dynamic calculation of transfer time between the soil P pools and initialization of different soil P pools with global data. Another improvement is the calculation of crop uptake using Michaelis-Menten kinetics, whereby the maximum uptake development reflects technology changes such as improved crop varieties.,

The global results of our study agree well with estimates from recent literature (Table 4). DPPS results from individual countries are also consistent with recent literature. Large P surpluses in China, India, Western Europe in our study agree with MacDonald et al. (2011), as well as P deficits in Argentina, the central parts of the United States and Eastern Europe (especially Kazakhstan). For most provinces in China, especially in eastern China, estimated P surpluses of > 10 kg P ha$^{-1}$yr$^{-1}$ are similar to both Shen et al. (2011) and MacDonald et al. (2011), who reported > 10 kg P ha$^{-1}$yr$^{-1}$ and > 13 kg P ha$^{-1}$yr$^{-1}$, respectively (Table 4).

In the early 20th century, the P uptake rates are only marginally smaller than the low P input rates, implying that P was balanced in crop production systems although producing low yields. P inputs have steadily increased globally since the 1950s, allowing for an increase in crop production, but ultimately leading to a decrease in the efficiency of P application and an increase in the soil P pool. Both our research and Sattari et al.(2012;2016) show that since 1980s, P application rates have been reduced in much of Europe while uptake continues to increase due to the supply of residual soil P.

Over the last couple of decades, P management has led to different P input rates, P pool sizes, and crop uptake rates for various regions throughout the globe. For example, the cropping area in Africa has increased by more than 40% in the past 4 decades, while both the labile and stable P pools are slightly decreasing due to negative budgets (Fig. 6). This indicates that in Africa there has been soil P depletion in cropland, and simultaneous expansion and probably also abandonment of arable land, replacing degraded soils by fresh forest or savanna.

In contrast, P surpluses in industrialized countries have been decreasing steadily from high values in the 1970s and 1980s towards a small deficit in recent years; a hysteresis effect of uptake is observed, with equal inputs resulting in low (1970s) and high (2010) P uptake, due to a gradual increase of labile pool over the whole period.

The large P surpluses in the industrialized countries in the 1970s and 1980s represent a legacy for future productive capacity. The present situation in many industrialized countries shows that the P fertilizer inputs and P surpluses can be reduced considerably and the PUE can be increased to high levels without a yield penalty. This phenomenon of the legacy of soil P due to large P surpluses, such as during the 1970s and 1980s in Western and Eastern Europe, the Russian Federation and the USA, has been recognized before (Sattari et al., 2012). The data for the transition countries show that this soil P depletion can continue for many years without yield declines. However, the inputs have been reduced to low levels and surpluses turned into deficits, and the current model can be eventually used to investigate how long this depletion can continue without a yield decline by P limitation.

China and India have shown large surpluses of Pin particular since the 1990s, and are currently building up large soil P reserves, but heterogeneously distributed with negative budgets in some parts. In 2010, the residual soil P in China and India is 18 and 17 kg P ha$^{-1}$ yr$^{-1}$, respectively, compared to 1 kg P ha$^{-1}$yr$^{-1}$ in the USA. The total accumulated residual P per hectare of cropland in China and India exceeds that in Western Europe, Russian Federation and the USA, suggesting that it may be possible to reduce P application rates without affecting crop uptake.



## 5 Conclusion

In this study we present a spatially explicit model-based inventory of global soil P stocks and crop uptake for the period of 1900-2010, which represents the time in the Anthropocene when human activities accelerated the global agricultural P cycle by more than a factor of 10. The model can simulate the crop P uptake and the various soil P pools. It matches historical P

uptake based on production data for cropland in most countries of the world.

Since accumulation of P is a localized process, by using the spatially explicit DPPS model, local pools can be matched to their uptake. We presented the long-term (> 100 years) simulation of changes in soil P pools and crop uptake with a high resolution (0.5 by 0.5 degree, rather than continental scale) while accounting for finer scale heterogeneity by including

within grid cell landuse change. Compared with previous DPPS application by Sattari et al. (2012), our research has improved DPPS in calculating the spatial variability of the soil P pool changes within countries by accounting for soil characteristics and dynamic parameters, calculating the process-based uptake rather than fixed value, considering the yearly landuse change, initializing different soil P pools with global data and modelling P losses by runoff instead of fixed numbers. These improvements make the spatial explicit DPPS model allow to analyze long-term plant-soil P interaction and contribute

to sustainable P management, plant nutrition and food security at the global scale.

Model results indicate that the global P inputs (including mineral fertilizer and manure) have increased from 2.0 Tg P yr$^{-1}$ in 1900 to 23.0 Tg P yr$^{-1}$ in 2010 with large variation across different regions and countries of the world. At the global scale, about half of the applied P in 2010 was taken up by harvested crops (modelled annual rates of 12 Tg P yr$^{-1}$, 7.3 kg P ha$^{-1}$;

data 13.8 Tg P yr$^{-1}$, 8 kg P ha$^{-1}$). All world regions and countries show similar patterns before 1950, i.e. very low P input levels and crop P uptake that are not very much different from inputs. However, after 1950, regions and countries show different soil P inputs, crop P uptake, soil P budgets and pool changes, which indicate the spatial and temporal variability of soil P condition.

According to the model sensitivity analysis, the maximum uptake level (*max_uptake*, reflecting the technology level), the initial labile pool size (*LP_yang*), the calibration coefficient which mimics the direct P availability within the labile pool (*fr_mobile*) and the slope of the P input response curve at the origin (*init_recovery*) have an important influence on crop P uptake. This means that the simulated crop P uptake is influenced by both soil properties and crop characteristics.

The simple DPPS model can be used to assess the long-term changes in the soil P status, an important indicator of soil fertility and future soil productivity. Via its gridded approach, this model can ultimately help to improve our mechanistic understanding of P cycle in global cropland at local and global scale and contribute to evaluate broad-scale food security and make sustainable P management policies.





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




**Table 1: The parameters of the DPPS model.**

| Parameter | Description (unit) | Method/Value/Source |
|---|---|---|
| fr_weathering | Fraction of apatite in soils that is released during weathering (no dimension) | 0.001747 |
| weathering | The annual amount of P that becomes available from the parent material (kg P ha$^{-1}$yr$^{-1}$) | fr_weathering multiplied by the soil P present as apatite according to Yang et al. (2010) |
| deposition | The annual amount of P from atmospheric deposition (kg ha$^{-1}$yr$^{-1}$) | Modelled gridded P deposition from Mahowald et al. (2008) |
| fertilizer | Fertilizer P input (kg ha$^{-1}$yr$^{-1}$) | Beusen et al. (2016) |
| manure | Manure P input (kg ha$^{-1}$yr$^{-1}$) | Beusen et al. (2016) |
| LP | The LP pool, calculated for every grid every year (kg ha$^{-1}$) | Equation. 1 |
| SP | The SP pool, calculated for every grid every year (kg ha$^{-1}$) | Equation. 2 |
| LP_yang | The initial labile pool size | Yang et al. (2010) |
| SP_yang | The initial stable pool size | Yang et al. (2010) |
| max_uptake | The maximum amount of uptake by plants from availableP (kg ha$^{-1}$yr$^{-1}$) | 100 |
| init_recovery | The initial recovery fraction, which is the slope of the P response curve at zero, specific per soil type (no dimension) | Batjes (2011) |
| fr_mobile | The calibration coefficient which mimics the direct P availability within the labile pool | Equation. 7 |
| availableP | The amount of P available to plant roots (kg ha$^{-1}$yr$^{-1}$) | fr_mobile × LP |
| litter | The P content in crop litter that returns to the soil (kg ha$^{-1}$yr$^{-1}$) | Soils under natural vegetation: see text<br>Arable soil: 0 |
| $\mu_{LS}$ | Transfer time for LP to SP (years) | 5 (Sattari et al., 2012) |
| $\mu_{SL}$ | Transfer time for SP to LP (years) | Equation. 5 |
| uptake | Crop P uptake (kg ha$^{-1}$yr$^{-1}$). | Soils under natural vegetation: the inputs from litter, deposition and weathering<br>Arable soil: Equation. 6 |
| agri_2_natural_years | The years for abandoned land to revert to natural conditions | 30 |





**Table 2: The parameters included in the sensitivity analysis.**

| Symbol | Min | Default | Max | Type |
|---|---|---|---|---|
| $\mu_{LS}$ | 3 | 5 | 7 | Range |
| fr_weathering | 0.0014 | 0.001747 | 0.0021 | Range |
| agri_2_natural_years | 10 | 30 | 50 | Range |
| max_uptake | 0.75 | 1 | 1.25 | Multiplier |
| fr_mobile | 0.75 | 1 | 1.25 | Multiplier |
| LP_yang | 0.75 | 1 | 1.25 | Multiplier |
| TP_yang | 0.75 | 1 | 1.25 | Multiplier |
| deposition | 0.75 | 1 | 1.25 | Multiplier |
| init_recovery | 0.75 | 1 | 1.25 | Multiplier |
| runoff | 0.75 | 1 | 1.25 | Multiplier |
| fertilizer | 0.75 | 1 | 1.25 | Multiplier |
| manure | 0.75 | 1 | 1.25 | Multiplier |



**Table 3: Standardized regression coefficient (*SRC*) representing the relative sensitivity of the initial (1900) *LP***

**and *SP* pools and crop P uptake representing global model results (columns) for the year 1950 and 2000 to**

**variation in 12 model parameters.**

| Parameter | LP | | SP | | uptake | |
|---|---|---|---|---|---|---|
| | 1950 | 2000 | 1950 | 2000 | 1950 | 2000 |
| $\mu_{LS}$ | 0.04 | 0.09 | 0.00 | -0.01 | | 0.05 |
| fr_weathering | 0.04 | 0.04 | 0.01 | 0.02 | | 0.04 |
| agri_2_natural_years | | | | | | |
| max_uptake | -0.07 | -0.12 | -0.03 | -0.05 | **0.85** | **0.47** |
| fr_mobile | -0.03 | -0.11 | -0.01 | -0.03 | **0.29** | **0.50** |
| LP_yang | **1.00** | **0.96** | -0.09 | -0.11 | **0.23** | **0.37** |
| TP_yang | -0.02 | -0.08 | **1.00** | **0.97** | - | -0.03 |
| deposition | | | | | | |
| init_recovery | -0.02 | -0.10 | -0.01 | -0.03 | **0.29** | **0.51** |
| runoff | | | | 0.00 | | |
| fertilizer | 0.06 | **0.24** | 0.02 | 0.10 | 0.03 | 0.18 |
| manure | 0.06 | 0.10 | 0.03 | 0.06 | 0.02 | 0.06 |

[a] Cells with no value represent insignificant *SRC* values; all cells with values have significant SRC, numbers with normal
font indicate values -0.2 < *SRC* < 0.2; numbers with bold font indicate values exceeding -0.2 and 0.2. An *SRC* value of 0.2
indicates that the parameter concerned has an influence of $0.2^2 = 0.04$ (4 %) on the model variable considered. We refer to
this as an important effect on the model output variable considered. The blank items are the ones that are not significant. The
full results with data for world regions and selected countries are in Table S4.



**Table 4: Comparison with other studies for global cropland.**

| Reference | P fertilizer application (Tg P yr$^{-1}$) (year of estimation) | Manure P application (Tg P yr$^{-1}$) | Crop P uptake (Tg P yr$^{-1}$) | Agronomic soil P budget (Tg P yr$^{-1}$) |
|---|---|---|---|---|
| Sattari et al. (2012) | 23.2 (2007) | | 11.5 | 11.7 |
| MacDonald et al. (2011) | 14.2 (2000) | 9.6 | 12.3 | 11.5 |
| Chen and Graedel (2016) | 20.2 (2013) | 6.5 | 12.4 | 14.3 |
| Bouwman et al.(2009) | >13 (2000) | 7.0 | 10.0 | 11.0 |
| This study | 13.6 (2000) | 6.1 | 10.7 | 9.0 |
| | 15.8 (2010) | 7.2 | 12.0 | 11.0 |

P fertilizer application is the application of mineral fertilizer; agronomic soil P budget is the difference between the sum of P mineral fertilizer and manure and crop P uptake (agronomic soil P budget = P fertilizer application + manure P application - crop P uptake).





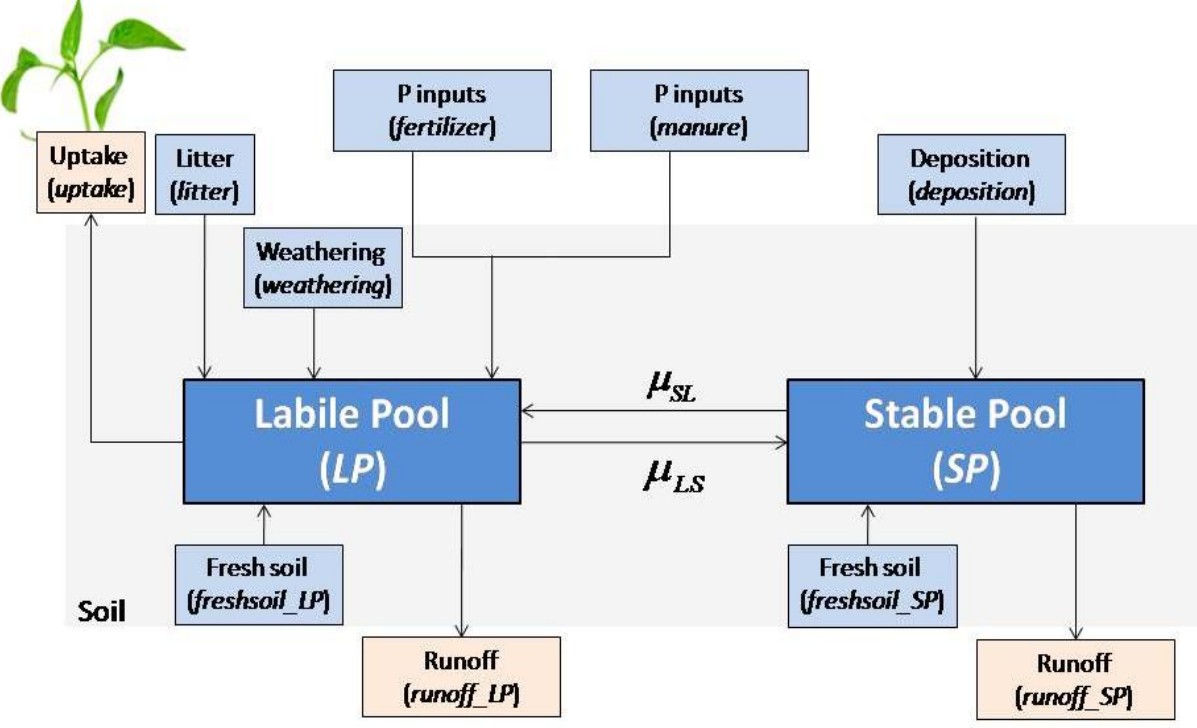

**Figure 1: Scheme of the DPPS model. The model includes two dynamic P pools, i.e. the labile pool (*LP*) and the stable pool (*SP*), comprising both of organic and inorganic P. Five inputs of P to the system are defined: mineral fertilizer and manure, weathering, deposition, fresh soil and litter. $\mu_{LS}$ and $\mu_{SL}$ (years) denote the transfer time of P from *LP* to *SP* and from *SP* to *LP*, respectively.**

5 **Modified from Sattari et al. (2012)**



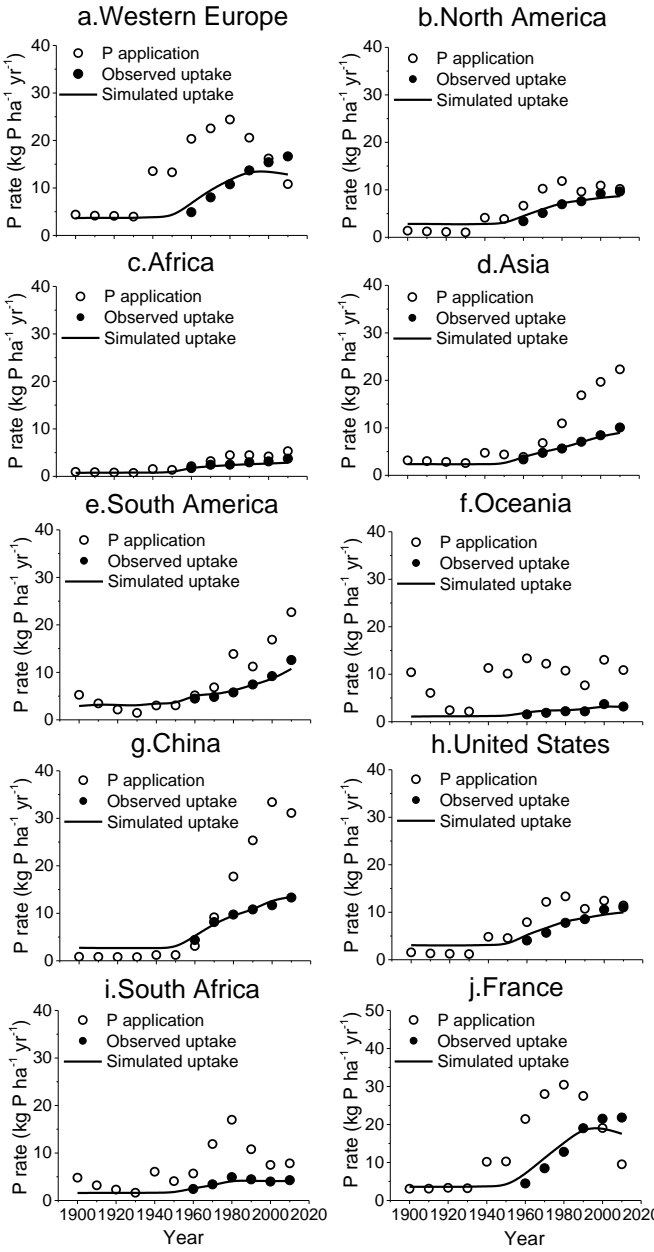

**Figure 2: Trends of annual P application (including manure and P fertilizer), historical P uptake and simulated P uptake in cropland for the period 1900-2010 in the continents and world regions : Western Europe (a) , North America (b) , Africa (c), Asia (d), South America (e) and Oceania (f), and the countries : China (g), United States (h), South Africa (i) and France (j). Open and**

5 **closed dots refer to P application and observed P uptake rates from FAO validation data. Solid lines refer to simulated P uptake rates. Western Europe includes Andorra, Austria, Belgium, Denmark, Faroe islands, Finland, France, Germany, Gibraltar, Greece, Holy See (Vatican City State), Iceland, Ireland, Italy, Liechtenstein, Luxembourg, Monaco, Netherlands, Norway, Portugal, San Marino, Spain, Svalbard and Jan Mayen, Sweden, Switzerland, United Kingdom.**




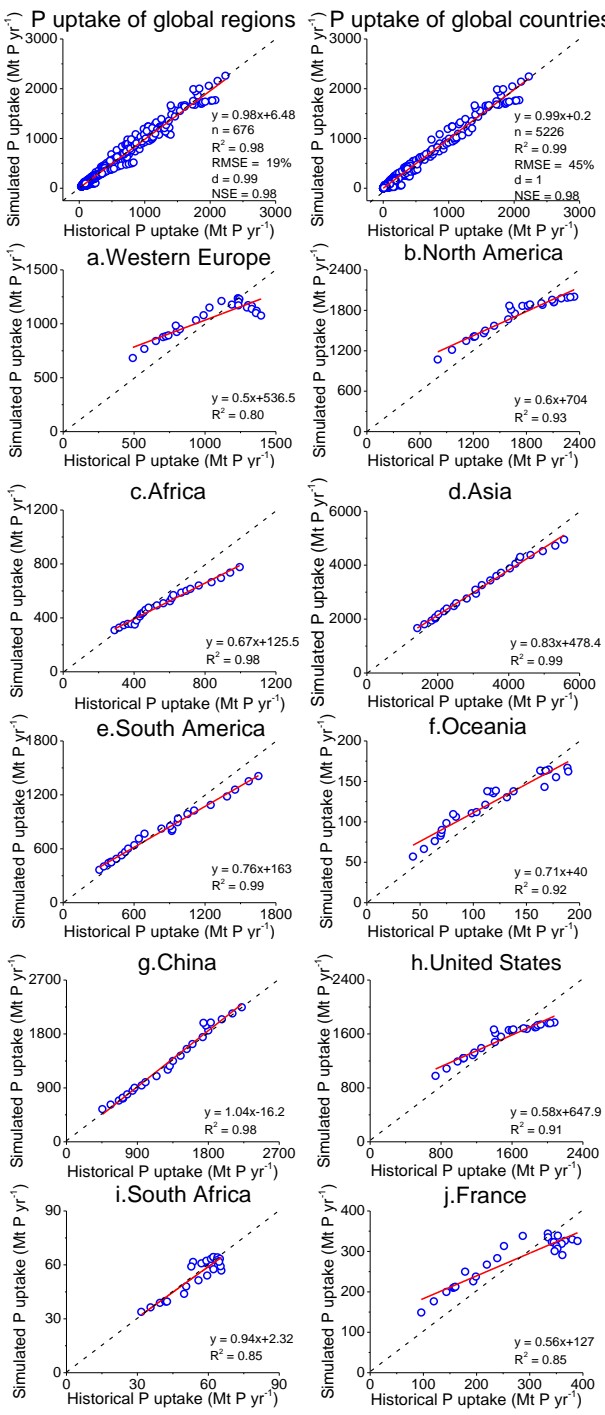

**Figure 3: Relationship between simulated and historical P uptake for individual regions, countries and global combined with the results of every two years from 1960 to 2010. Dashed line is 1:1**





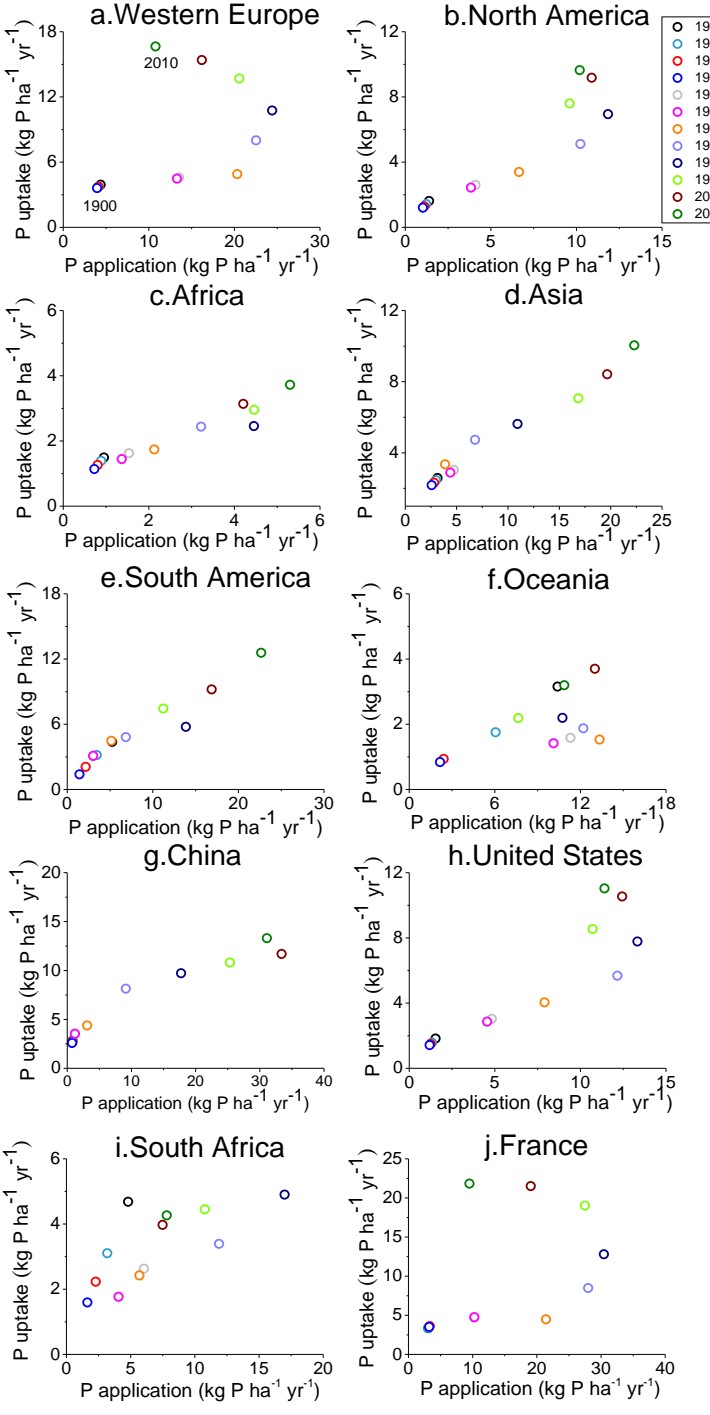

**Figure 4: P uptake vs. P application between 1900 and 2010. There are two P uptake rates with the same amount of P application, which shows the contribution of residual P to the crop production in Western Europe (a) and France (j).**




**Figure 5: The total soil P inputs (fertilizer, manure and deposition), soil P outputs(crop uptake and runoff) and biogeochemical soil P budget (soil P input - soil P output) of different regions and countries. Soil P inputs are positive and outputs are negative.**



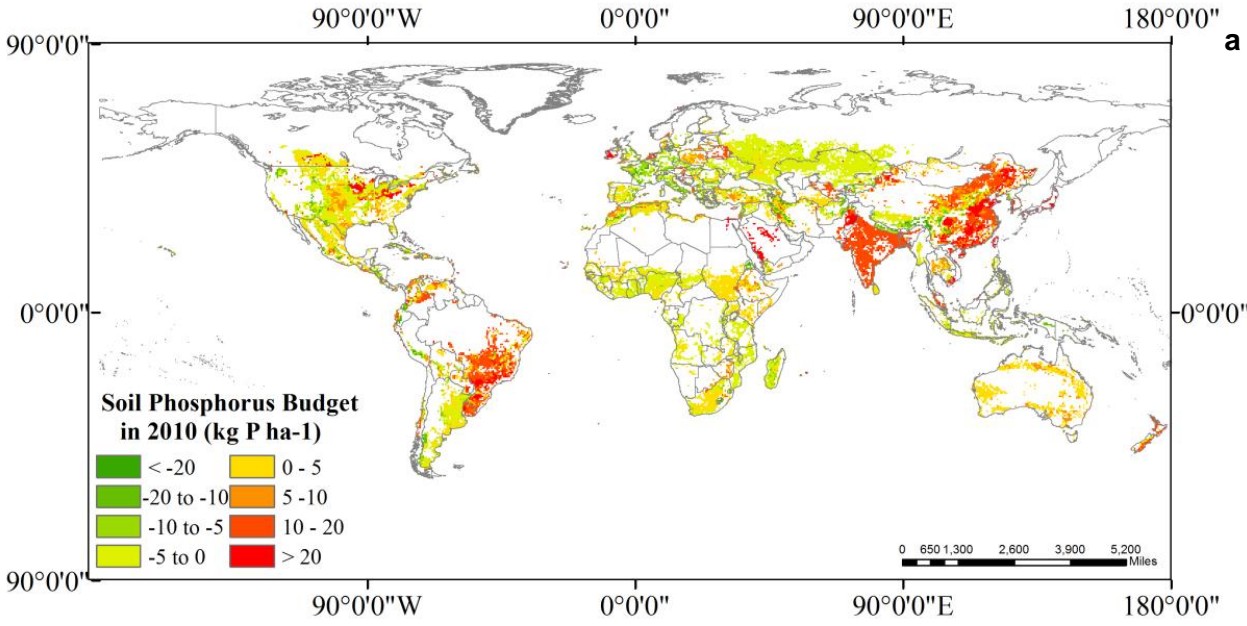

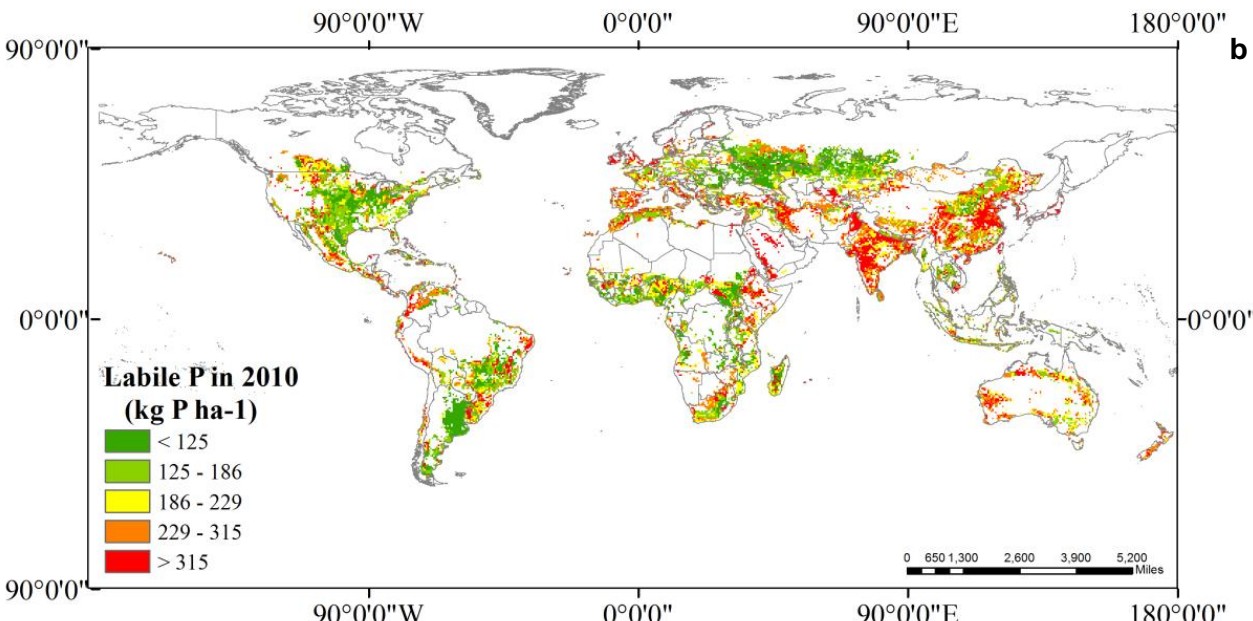





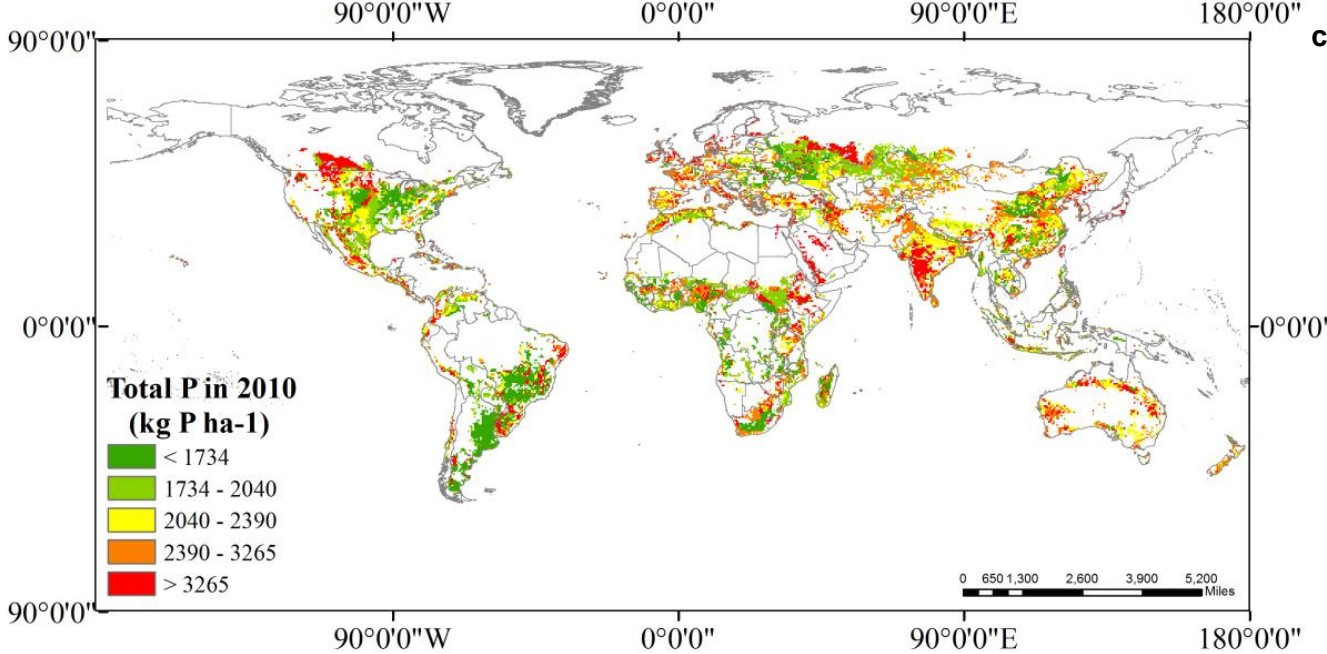

**Figure 6: (a) The figure of soil P budget in 2010. Soil P budget is the difference between total soil P input (mineral fertilizer, manure and deposition) and total soil P output (uptake and runoff); (b) and (c) the global distribution of labile and total P in global agricultural land soils in 2010. Yearly grid maps for the period 1900-2010 can be viewed in supplementary movie S5.**



**Figure 7: Changes of soil *LP* (a and b) and *TP* pools (c and d) in arable land for different regions and countries for the period 1900-2010.**

