# Peer review of "Spatiotemporal dynamics of soil phosphorus and crop uptake in global cropland during the twentieth century"

_Biogeosciences, 2016_

## Referee Comment (RC1) · Anonymous Referee #1 · 24 Jan 2017

This manuscript studied the spatial spatiotemporal dynamics of soil phosphorus and crop uptake in global cropland by using a simple mass balance model. Highlights of this paper is the large spatial scale and long temporal scale. However, there are some problems in this paper that should be carefully reconsidered by authors. 1) There exists temporal scale gap between crop uptake to phosphorus and the simulated results from this model. The time that crop uptakes phosphorus is in several months but the temporal scale from the model is ten years (one century but ten years step). In other words, the simple mass balance model used in this paper was not fully demonstrated in calculating the phosphorus untaken by crop. And this should be explained in the introduction. 2) About model's construction and veracity. This is a simple model that is

based on mass balance. It is important to study about the soil P cycle and plant cycle. The model fails to consider external phosphorus input from other land-use types like forest land or grass land when these types were converted to cropland. 3) There should have more words to explain the part about soil P loss and how to calculate this part in this paper. Soil P loss is a key role in the entire phosphorus cycle. 4) Something about the improvement about the original model. The improvement on the original model is not clear. In my opinion, the authors just changed the input data at world scale into country scale. If not, please describe them and tell readers what your innovation in detail is. 5) Precipitation and temperature are two key parameters in phosphorus cycle, and these two parameters have significant influences on the transformation between the two phosphorus pools. 6) There are many crop types in this world, and their capacity to uptake phosphors are very different. Please address this problem in detail. 7) There are many driving factors changing the spatial spatiotemporal dynamics of soil phosphorus and crop uptake in global cropland, like changing landuse and increasing input of artificial phosphorus. These factors should be discussed and the most key factors should be identified. 8) The equation 7 is not easily understood by the readers. Please clarify it. 9) There are many similar researches at the world scale and many researches have more longer temporal scale than this paper? So, please clearly clarify the innovation points of this paper.

---

## Referee Comment (RC2) · W. Shangguan (Referee) · 15 Feb 2017

This work developed a spatial explicit DPPS (Dynamic Phosphorus Pool Simulator) model based on Sattari et al., 2012. Except spatial explicit calculation, improvements include the incorporation of land use change, dynamic transfer from SP to LP, initialization P pools with global observations, dynamic P loss by runoff, use of Michaelis-Menten kinetics in the calculation of crop uptake, and time-variant maximum uptake parameter. Most improvements are based on previous studies. However, major findings from the model results are already made by previous studies such as MacDonald et al, 2011, Sattari et al., 2012, Bouwman et al 2013. The authors need to address the new findings clearly. Major comments: My major concern is the setting of max_updake.

The reason of setting a time-variant max_update is good, i.e. the development of technology in crop production. The authors used the historical uptake, but no reference and data period was given. I do not believe this change is abrupt but gradual, and the second agriculture revolution happened between 1930-1960. It is better to set a max_updake curve according to crop yield per area or other index, instead of two arbitrary values. In addition, why max_update will change for no cropland? As the uptake is quite sensitive to max_updake (table 3), I believe that the modeling results depends heavily on the setting of max_updake. And I hope there may be some new findings.

My second major concern is that it seems that the authors used the same uptake dataset (1960-2010) to calibrate and validate the model. To evaluate the model performance, the validation data should be independent from the calibration. May be use only data from 1960-2000 to calibrate and use data from 2000-2010 to validate. But you could use all the observation data in calibration to show other results after the evaluation. Furthermore, there is no observation from 1900-1960, so the model estimation for this period is a kind of reconstruction, which has higher uncertainty. This should be emphasized in the text.

Improvements of the manuscript structure should be made to make it more readable. Here are some of my suggestion. As this paper is an improvement of DPPS, you may descript the improvements in the introduction, which can make it more clear to the readers in the first impression. Move content from page 5 line 16~20 to the section 2.3. Add section 2.4 validation, and move page 5 line 16, page 6 line 21~27 to this section. Maybe add a section called initialization and calibration before section 2.3, which contains page 5 line 22~30. There are no description or interpretation of table 3 in the results or discussion.

I am confused by the two separation of the simulated 100 years: 1950 and 1960. Some parameters and calculations are set by 1960 split and some by 1950 split. Why not just set the same time split? If you choose one, why do you select 1950 or 1960? I understand that the uptake observations are from 1960, but it should not be used as a

criterion in time split of parameter setting.

What is the advantage of initialization by global observations? Does it speed up the simulation or improve the model results? What is the effect of land use change? Add a figure to show this. In the discussion, the difference with previous studies, especially spatially explicit, should be addressed and explained the possible reasons.

Minor comments: Table 1: the value of max_uptake is not 100 Figure 1: Add features of the new DPPS in the figure: land use change, spatial explicit and initialization with global data. Why change the P inputs without entering the stable pool compared to Sattari et al, 2012? Figure 3: no definition of d and NSE. Describe and explain the underestimation and overestimation in different regions. Figure 4: color are too hard to distinguish, may be use also symbols like cross. Page 3, line 20: at 30 cm over? Page 5, line 16: Due to the equation in S1, it is not RMSE but Normalized RMSE (i.e. coefficient of variance). Please correct it. Page 10, line 26: change Pin to P in

---

## Author Comment (AC1) · 8 Mar 2017

**Spatiotemporal dynamics of soil phosphorus and crop uptake in global cropland during the twentieth century**

Jie Zhang[1], Arthur H.W.Beusen[2,3], Dirk F. Van Apeldoorn[2,4], José M. Mogollón[2], Chaoqing Yu[1], Alexander F. Bouwman[2,3]

[1]Department of Earth System Science, Tsinghua University, Beijing, 100084, China
[2]Department of Earth Sciences-Geosciences, Faculty of Geosciences, Utrecht University, P.O. Box 80021, 3508 TA Utrecht, The Netherlands
[3]PBL Netherlands Environmental Assessment Agency, P.O. Box 30314, 2500 GH The Hague, The Netherlands
[4]Farming Systems Ecology, Wageningen University, P.O. Box 430, 6700 AK Wageningen, The Netherlands

**Response to the reviewers**

We are very grateful to the two reviewers for their constructive feedback. The suggestions definitely led to significant improvements of our text, particularly where the reviewers found sections that were not clear enough. The concerns have been addressed below and in the revised manuscript. Below are the reviewer comments, **our response in bold.**

**Reviewer 1**
1) There exists temporal scale gap between crop uptake to phosphorus and the simulated results from this model. The time that crop uptakes phosphorus is in several months but the temporal scale from the model is ten years (one century but ten years step). In other words, the simple mass balance model used in this paper was not fully demonstrated in calculating the phosphorus untaken by crop. And this should be explained in the introduction.

**This model uses an annual time step. The results are merely plotted every ten years in order to span the entire 1900-2010 timeframe without overcrowding the figures. The crop phosphorus (P) uptake is plotted every four years in the supplemented movie. This is specified in the first paragraph of section 2.1 Model Description.**

2) About model's construction and veracity. This is a simple model that is based on mass balance. It is important to study about the soil P cycle and plant cycle. The model fails to consider external phosphorus input from other land-use types like forest land or grass land when these types were converted to cropland.

**Yearly land use change within a grid-cell (e.g cropland expansion and cropland**

abandonment) is provided by the IMAGE model with a 0.5 by 0.5 degree resolution. We assume that yearly cropland expansion within a cell is initiated with new pool of virgin soil without fertilizer history according to Yang et al. (2013). For land abandonment (arable land to natural land) we assume that it takes 30 years for abandoned land to revert to natural conditions, and in this period the P in *litter* increases and *uptake* decreases linearly with time from zero to the natural flux. For further clarification, the following text has been added/modified to the manuscript:

"In grid cells where cropland expansion takes place, the initial conditions of virgin soil (without fertilizer history) are assigned to the new, additional area in the year considered. For land abandonment (arable land to natural land) we assume that it takes 30 years for abandoned land to revert to natural conditions, and in this period the P in *litter* increases and *uptake* decreases linearly with time from zero to the natural flux."

3) There should have more words to explain the part about soil P loss and how to calculate this part in this paper. Soil P loss is a key role in the entire phosphorus cycle.

We see that the description of the runoff calculations were not complete and unclear. We added a more detailed description, see lines 106-113. The description now reads:

"The global spatially explicit P runoff is calculated following the approach of Beusen et al. (2015) who distinguished losses from recent nutrient applications in the form of fertilizer, manure or organic matter (Hart et al., 2004), and a "memory" effect related to long-term historical changes in soil nutrient inventories (McDowell and Sharpley, 2001; Tarkalson and Mikkelsen, 2004). The memory effect is based on Cerdan et al. (2010) using slope, soil texture and land cover type to estimate country aggregated soil-loss rates for arable land, grassland and natural vegetation. The P content of the soil loss (*runoff_LP* and *runoff_SP*) is based on the *LP* and *SP* of the soil in the grid cell considered at that moment in time."

4) Something about the improvement about the original model. The improvement on the original model is not clear. In my opinion, the authors just changed the input data at world scale into country scale. If not, please describe them and tell readers what your innovation in detail is.

Although the two models are developed follow the same philosophy, the spatial explicit DPPS model has been improved in several aspects. These have been listed in Figure 1, and furthermore expanded upon in new Table 1. We have also added the following text to the Introduction:

"Compared with previous studies, our model has been improved to include spatially explicit calculations, land use change, dynamic transfer between different soil P pools, initialization of the P pools with global observations,

**dynamic P loss by runoff, dynamic calculation of crop P uptake and time-variant maximum uptake parameter (Table 1)."**

Table 1. Comparison between non-spatial DPPS (Sattari et al., 2012) and the spatially explicit DPPS model.

| Property | Non-spatial DPPS model (Sattari et al., 2012) | Spatially explicit model |
|---|---|---|
| Spatial Resolution | Continental | 0.5 degree |
| Land use changes | No | Considering arable land expansion |
| Variable budget of soil P | Fixed | Recalculated every year |
| Initialization of soil pools | No | Initialized the soil pools for the pre-industrial period |
| Crop P uptake | Fixed fraction | Michaelis-Menten kinetics |
| P flow ($\mu LS$ and $\mu SL$) | Fixed number | Calculated for every grid cell |
| Erosion and runoff | Fixed number | Changed with the pool size |
| Soil properties | One class in continental level | Grid-based resolution |

5) Precipitation and temperature are two key parameters in phosphorus cycle, and these two parameters have significant influences on the transformation between the two phosphorus pools.

**Precipitation and temperature are implicitly included in the biogeochemical background of P distribution for natural soils (e.g. the initial distributions of LP and SP from Yang et al., 2013, are a direct consequence of climate) and other input datasets for our model, such as weathering, deposition, and runoff. While precipitation and temperature may be important for the transformation between different forms of phosphorus, they play a secondary role with respect to other factors such as the farming practices of crop production and fertilizer inputs. We have furthermore now added a reference to a study that compared the various parameters affecting P distribution in soils:**
**"Ringeval et al. (2017) qualified the contribution of different factors to the global soil P distribution and found that soil biogeochemical background and farming practices are the most important drivers for P uptake. This agrees with our focus on these factors as the biggest controllers for P distribution in cropland soils. The P content of soils under natural vegetation and the $CaCO_3$ content, pH, clay, moisture and water content, determines the capacity for P sorption, retention, and soil weathering. For example, in arid and semi-arid regions the low soil water content hinders P diffusion and plant root growth, ultimately limiting crop P uptake. While this chemical background is highly heterogeneous, our gridded**

**approach and model initialization capture some of these differences in global soils by distributing LP and SP according to natural soils of Yang et al. (2013)."**

**Downscaling to a daily time step to capture precipitation variations is outside the scope of this model and would require additional parameterization for these processes. In any case, the point of this work is to show the spatiotemporal dynamics of soil P reserves and crop uptake for the entire globe at centennial scales, and the impacts of fertilizer on soil P and crop uptake.**

6) There are many crop types in this world, and their capacity to uptake phosphors are very different. Please address this problem in detail.

**We collected all (>120) annual and perennial food, feed and fodder crops and fruits from FAO statistics and aggregated them into 34 crop groups as distinguished in recent FAO studies (Alexandratos and Bruinsma, 2012). P content for each crop group was illustrated in Table S3 based on different sources (Bouwman et al., 2005) and the crop P uptake was calculated as crop yields multiply P content of this crop group. This is directly mentioned in the Data Used subsection:**
**"For the period 1960 to 2010, simulated crop P uptake is validated against P in harvested crop production data covering all (>120) annual and perennial food, feed and fodder crops and fruits from FAO statistics (FAO, 2016a, b) and are aggregated into 34 crop groups distinguished in recent FAO studies (Alexandratos and Bruinsma, 2012). The historical P uptake per hectare is calculated from the crop yields from the above-mentioned FAO statistics for 34 crop groups and P content for each crop group, as listed in Table S3."**

7) There are many driving factors changing the spatial spatiotemporal dynamics of soil phosphorus and crop uptake in global cropland, like changing landuse and increasing input of artificial phosphorus. These factors should be discussed and the most key factors should be identified.

**The input of artificial phosphorus fertilizer and manure has been thoroughly discussed throughout the text. The driving factors for P dynamics in croplands have also been discussed in the text. We have, however, added a final paragraph to the Discussion addressing the global impact of land use change:**
**"The contribution of expanding cropland areas and crop yields (P uptake per hectare) to the production increase since 1960 is about 18% for the former and 82% for the latter. Nevertheless, there are likely regional differences with developing countries having a greater proportion of agricultural areal expansion and industrialized countries more improvements in nutrient use efficiency (Figure 4). This explains why for example in Africa, the labile and total P pools per hectare have been relatively constant, while uptake has exceeded inputs during the whole period 1900-2010."**

8) The equation 7 is not easily understood by the readers. Please clarify it.

**We agree that the original text was not clear enough. The calibration procedure is now explained more clearly in a separate section 2.3.**

9) There are many similar researches at the world scale and many researchers have more longer temporal scale than this paper? So, please clearly clarify the innovation points of this paper.

**We have now clearly summarized the improvement of this model compared to the original DPPS in Table 1 and in the text. Regarding other studies we have compared our results with recent studies, in order to show that all these models show results that are in the same order of magnitude. Unfortunately, it is not possible to compare our results for the changes in soil P pools, since all the studies presented in Table 5 have no inventories of changes in soil P reserves. This has now been stated in the Discussion.**

**Reviewer 2**

1. This work developed a spatial explicit DPPS (Dynamic Phosphorus Pool Simulator) model based on Sattari et al., 2012. Except spatial explicit calculation, improvements include the incorporation of land use change, dynamic transfer from SP to LP, initialization P pools with global observations, dynamic P loss by runoff, use of Michaelis-Menten kinetics in the calculation of crop uptake, and time-variant maximum uptake parameter. Most improvements are based on previous studies. However, major findings from the model results are already made by previous studies such as MacDonald et al, 2011, Sattari et al., 2012, Bouwman et al 2013. The authors need to address the new findings clearly.

**This comment is a combination of comment #4 and comment #9 by reviewer 1. In summary, we added text to the Introduction and Table 1 that summarizes the differences with Sattari et al. (2012). In the discussion we present the results of our models in comparison with a number of recent other studies. The major point we stress is that we cannot compare the dynamic of global soil P pools with other studies, because there is none with estimates of soil P reserves and changes therein (see Discussion).**

2. Major comments: My major concern is the setting of max_uptake. The reason of setting a time-variant max_uptake is good, i.e. the development of technology in crop production. The authors used the historical uptake, but no reference and data period was given. I do not believe this change is abrupt but gradual, and the second agriculture revolution happened between 1930-1960. It is better to set a max_uptake curve according to crop yield per area or other index, instead of two arbitrary values.

In addition, why max_uptake will change for no cropland? As the uptake is quite sensitive to max_uptake (table 3), I believe that the modeling results depends heavily on the setting of max_uptake. And I hope there may be some new findings.

**We agree with the reviewer that *max_uptake* should be related to crop yield. Indeed, crop yield as expressed by the uptake is what is now used to calculate the *max_uptake*. We split the data period in two parts, prior to 1960 and after 1961. Actually, the split year should be somewhere between 1950 and 1960, which is the period during which fertilizer use actually started to increase rapidly, and so did the crop yields in most countries. Apparently our text was not clear so we have modified it to the following:**
**"Max_uptake is a time dependent variable, considering the development of technology in crop production. A first condition is that max_uptake never decreases. A second condition is that max_uptake has a maximum value of 100 kg P ha-1 yr-1, which exceeds the present-day highest value for all countries of the world based on our data (section 2.2). We make a split in 1961, the year in which the FAO data series start (FAO, 2016a, b). Prior to 1961 max_uptake is the minimum of twice the uptake calculated from crop production data for 1960, and the sum of P inputs from fertilizers and manure; from 1961 onwards, it is the uptake calculated from crop production data times a factor of 2, or the previous max_uptake if that is larger."**

**The sensitivity analysis shows that *max_uptake* is among the most important parameters in the modelled uptake, with *fr_mobile*, the *LP* initial pool size and the initial recovery fraction.**

3. My second major concern is that it seems that the authors used the same uptake dataset (1960-2010) to calibrate and validate the model. To evaluate the model performance, the validation data should be independent from the calibration. May be use only data from 1960-2000 to calibrate and use data from 2000-2010 to validate. But you could use all the observation data in calibration to show other results after the evaluation. Furthermore, there is no observation from 1900-1960, so the model estimation for this period is a kind of reconstruction, which has higher uncertainty. This should be emphasized in the text.

**Indeed, there is no historical record of crop P uptake for the period of 1900 to 1960. To circumvent this, we used the P inventory from a recent study, which was based on various P inputs into soil (Bouwman et al., 2013) to force the model. Consequently, this period has higher uncertainty. This is stated in Section 2.2:**

**"In this database, for the period 1900-1960, the gridded inventories of P inputs from a recent study (Bouwman et al., 2013) were used for the calculation of the soil P pools and crop uptake. This dataset was based on various sources for land use (Klein Goldewijk et al., 2011; Klein Goldewijk et al., 2010), livestock for 1900-1960 (Mitchell, 1998, 1993a, b), animal and crop production and fertilizer**

use for 1930-1950 (FAO, 1951) and fertilizer use prior to 1930 (Cressy Morrison, 1937), which were spatially distributed with IMAGE-GNM as described by Bouwman et al. (2013). Due to the lack of data during this time period, uptake estimates are more uncertain than those for the years since 1961 to 2010 based on FAO statistics (see below)."

**The purpose of the study was to calculate the interactions between various phosphorus pools in agricultural soils, and to downscale uptake results on a grid basis, not to calculate the countrywide uptake. The DPPS model is thus used here not as a forecasting model but rather as a hindcasting model to calculate the historical (up to 2010) interactions and changes in the LP and SP soil pools, assuming that we can reasonable estimate the known uptake at the country, regional, and global scales (hence the calibration). Soil P measurements are very heterogeneous at the 0.5x0.5 degree scale. Our results thus represent the bulk average for all the agricultural lands in these cells. There is no data that we can use to validate the LP and SP contents at this scale and one of the reasons that these types of models are important for the global phosphorus cycle.**

4. Improvements of the manuscript structure should be made to make it more readable. Here are some of my suggestion. As this paper is an improvement of DPPS, you may descript the improvements in the introduction, which can make it more clear to the readers in the first impression. Move content from page 5 line 16_20 to the section 2.3. Add section 2.4 validation, and move page 5 line 16, page 6 line 21_27 to this section. Maybe add a section called initialization and calibration before section 2.3, which contains page 5 line 22_30. There are no description or interpretation of table 3 in the results or discussion.

**We already discuss the model improvements under reviewer 1 #9. We added a new subsection "Initialization and Calibration". Table 3 is described in Section 2.3 (Sensitivity Analysis) and we added more interpretation in the new version. We do not include a section on validation, as this model is not independently validated with measured data.**

5. I am confused by the two separation of the simulated 100 years: 1950 and 1960. Some parameters and calculations are set by 1960 split and some by 1950 split. Why not just set the same time split? If you choose one, why do you select 1950 or 1960? I understand that the uptake observations are from 1960, but it should not be used as a criterion in time split of parameter setting.

**The is no real 1950 split, only 1960/1961 is used as a split (see comment #3 reviewer 2). In the discussion, we use 1950, which is actually the year when global fertilizer use actually started to increase dramatically. We use 1960/1961**

**as a split for *max_uptake* because this is the year when FAO statistics time series begin.**

6. What is the advantage of initialization by global observations? Does it speed up the simulation or improve the model results? What is the effect of land use change? Add a figure to show this. In the discussion, the difference with previous studies, especially spatially explicit, should be addressed and explained the possible reasons.

**As this model represents a forward run for P dynamics and storage in agricultural soils, we require initial values for the data. Initializing in a spatially explicit way allows for capturing heterogeneities in agricultural systems within a country, something that cannot be obtained from FAOSTAT countrywide data alone. We initialize the model with the global dataset from Yang et al. (2013), which to our knowledge, is the only published data providing the spatially explicit information about the global distribution of different forms of P in natural soils.**

**When the cropland expansion happens, natural soil with LP and SP contents from Yang et al. (2013) becomes agricultural soil and becomes altered due to agricultural practices. Soils start to receive the anthropogenic fertilizer (including mineral fertilizer and animal manure) in order to increase crop uptake, which also changes the amount and spatial distribution of different forms P in soils. The initial conditions are assigned to each new, additional area added to cropland each year. For land abandonment (arable land to natural land) we assume that it takes 30 years for abandoned land to revert to natural conditions, and in this period the P in litter increases and uptake decreases linearly with time from zero to the natural flux.**

**We have added a paragraph that discusses the fraction of area expansion vs. yield increase on the P crop total production on a global scale (see comment #7, reviewer 1).**

**For a comparison with other studies we refer to our response to reviewer 1, comment #9.**

7. Minor comments:

Table 1: the value of max_uptake is not 100

**The maximum possible value of *max_uptake* is 100 kg P ha$^{-1}$**

Figure 1: Add features of the new DPPS in the figure: land use change, spatial explicit and initialization with global data.

**We have added these changes to Figure 1 and have added a new Table 1 summarizing the new features of the model.**

Why change the P inputs without entering the stable pool compared to Sattari et al, 2012?

**We believe the current distribution of fertilizer and manure toward the LP and SP pools better represents reality as compared to Sattari et al. (2012).**

Figure 3: no definition of d and NSE.

**We have excluded d and NSE in the revision, to avoid confusion.**

Describe and explain the underestimation and overestimation in different regions.

**We added a paragraph on the model deviations in the discussion section:**

**"The reason for P uptake underestimation in regions like Western Europe in recent years is that the model is calibrated using fr_mobile over the whole period 1961-2010. Hence, the results for the middle of this period are very close to observed uptake values, but in the early and late part of the period 1961-2010, the model may deviate from the observations. We have deliberately chosen for a fixed value of fr_mobile. In theory, fr_mobile could be estimated by using the characteristics of the different fertilizers used, but such data is unavailable at our gridded scale. Furthermore, estimating a time-varying parameter describing P availability would be extremely complex and falls outside the scope of this study."**

Figure 4: color are too hard to distinguish, may be use also symbols like cross.

**In the revision we will include a figure with symbols (see uploaded figure 4).**

Page 3, line 20: at 30 cm over?

**We have modified the text according to the reviewer's suggestion.**

Page 5, line 16: Due to the equation in S1, it is not RMSE but Normalized RMSE (i.e. coefficient of variance). Please correct it.

**We have modified the text according to the reviewer's suggestion.**

Page 10, line 26: change Pin to P in

**We have modified the text according to the reviewer's suggestion.**